# Female Scent Activated Expression of Arginase1 and Inducible NO-Synthetase in Lung of BALB/c Male Mice

**DOI:** 10.3390/ani11061756

**Published:** 2021-06-11

**Authors:** Olga V. Gvozdeva, Kseniya M. Achasova, Nadezhda A. Litvinova, Elena N. Kozhevnikova, Ekaterina A. Litvinova

**Affiliations:** 1Siberian Federal Scientific Centre of Agro-BioTechnologies of the Russian Academy of Sciences, P.O. Box 267, 630501 Krasnoobsk, Russia; odostovalova@gmail.com (O.V.G.); achasovaks707@gmail.com (K.M.A.); e_zeste@yahoo.com (E.N.K.); 2Faculty of General Medicine, Kemerovo State Medical University, 650001 Kemerovo, Russia; nadyakemsu@mail.ru; 3Scientific Research Institute of Neurosciences and Medicine, 630117 Novosibirsk, Russia

**Keywords:** arginase1, inducible NO-synthetase, female scent, barrier immunity, olfactory search, breeding behavior, macrophages

## Abstract

**Simple Summary:**

The scent of receptive females stimulates the breeding behavior of male mice. However, searching for breeding partners is associated with an increased risk of sniffing contaminated fecal and urinary marks containing various infectious agents, activating males’ immunity to protect them from infection. We found that the scent of specific pathogen-free (SPF) females stimulated male mouse immunity but was not associated with changes in the hormonal background. The percentage of T-cells decreased after the female scent treatment, affecting a specific immune reaction. The female scent caused an activation of the *Nos2* and *Arg1* genes in the macrophages in the lungs of male mice, which could increase resistance to potentially infectious agents. Our data showed activation of nonspecific immunity in male mice in response to the sniffing SPF female scent. We suggest that our study may impact future multidisciplinary behavioral and immunological studies and veterinary practice.

**Abstract:**

Scent signals play an important role in the life of rodents. The scent of the opposite sex can modulate immunity. In mice populations with natural specific pathogens, in males, the scent of a female leads to a redistribution of leukocytes between the lung and the blood, resistance to the influenza virus, and a decrease in antibody production, but not in the development of inflammation induced by bacterial endotoxins. This study demonstrates the effect of the scent of soiled bedding of specific pathogen-free (SPF) status female mice on the percentage of different types of leukocytes in the blood, the expression of *Nos2*, *Arg1*, and *Foxp3* genes, and the presence of M1/M2 macrophages in the lungs of male BALB/c mice. The scent of the female SPF mice caused a redistribution between T- and B-cells in the blood, the increase in the expression of *Nos2*, *Arg1* genes, and the percentage of M1 type macrophages in the lung, but did not affect the different types of T-cells in the periphery or the lungs. Activation of macrophages in the lung is part of mucosal immunity, which is necessary for males as an adaptive mechanism to prevent potential infection during the search for a sexual partner.

## 1. Introduction

Chemosignals play a huge role in rodents’ lives, from finding food to avoiding predators [1]. Olfactory signals are involved in the reproductive physiology and sexual behavior of rodents. The scent of mature individuals can affect the puberty of juvenile animals and the ovulation of females, motivate the search for a sexual partner, and enhance the process of territorial marking in males [2,3,4].

A number of studies have shown that chemosignals of sexually mature conventional status females affect the immunity of male laboratory mice that have the same pathogens, causing the suppression of antibody production [5,6,7] and the activation of nonspecific immunity, such as leukocyte migration into the lungs, has been observed [8,9]. All mucous layers, including those of the upper respiratory tract and lungs have a nonspecific immune defense that includes antimicrobial proteins and immunoglobulins. Under the mucous layer, along with epithelial cells, immune cells are present [10]. Antigen-presenting cells (APCs) such as macrophages and dendritic cells are highly represented under the mucous layer. APCs meet pathogen antigens and present them to lymphocytes [11]. APCs, along with epithelial cells, secrete chemoattractants and cytokines that attract leukocytes to eliminate pathogens effectively [12]. For example, female-scent-induced migration of immunocompetent cells into the lungs increases mice’s resistance to the influenza virus, which may protect them from exogenous infection. The risk of catching an exogenous infection increases while searching for a sexual partner [13]. This phenomenon has been known for a long time; however, its mechanism is still unclear. The effect of the female mice scent on males’ immunity may be associated with specific pathogens of conventional status. The presence of pathogens has a significant effect on immune–endocrine interactions [14]. Whether the scent of females and non-pathogenic microbial patterns (SPF status mice) in soiled bedding affects the immunity of male mice has not been studied.

The rodent perception of scent signals is mediated by the main olfactory epithelium and the vomeronasal organ [15]. There are various receptors on the neurons of these organs that bind odorant and pheromonal molecules. Pheromone-sensitive vomeronasal receptors [16,17,18,19,20,21] and formyl-peptide receptors, which bind bacterial formyl-peptides and antimicrobial peptides, are located on the vomeronasal epithelium [22,23]. Olfactory receptors that perceive various odor molecules [24] and receptors for trace amines are located on the main olfactory epithelium [25]. Female soiled bedding that includes odor, pheromones, and non-pathogenic microbial patterns could suggest different mechanisms for the effect on male immunity. Firstly, this may be the result of the activation of the hypothalamic–pituitary system by the female scent. As a result of activation of the hypothalamic–pituitary system, the animal’s endocrine status can change, and as a result, steroid hormones affect the immune system [26,27,28,29,30,31]. Secondly, as has recently become known, there are receptors for odor molecules on immune cells [32,33,34,35] and, through these cells, the activation of immune cells and further effects on the immune system are possible.

Moreover, non-pathogenic microbial patterns can act on the receptors of dendritic cells and affect immunity. Studies of the phenomenon of immunity modulation by female scent (urine or contaminated bedding) were carried out using the scent of sexually mature females with specific pathogens (conventional status). The chemosignals of mature conventional status females activate the migration of neutrophils, eosinophils, and macrophages into the lungs [8,9] and reduce the number of antibody-forming spleen cells and the level of specific antibodies in the blood [5,6,7]. This immunomodulatory effect of the female scent increases resistance to viruses that the animal could potentially catch during the olfactory search [13]. However, how the SPF female scent affects immune cells—lymphocytes and macrophages—in the lungs remain unstudied. In this work, we have determined the expression level of the main metabolites and receptors of macrophages: arginase1 and CD209 (DC-IGN) for the M2 type and inducible NO-synthetases and CD86 for the M1 type. An effective M1 macrophage response is associated with resistance to bacterial infections. M2 macrophages realize anti-inflammatory and reparation responses. Despite the fact that M1 and M2 macrophages perform different functions and exhibit different phenotypes, both types of macrophages co-express many genes; however, some M1- and M2-specific genes also exist. These genes are markers for macrophages. For M1 macrophages, the classical marker is the *Nos2* gene; for M2 macrophages—the arginase1 gene. Moreover, we determined the level of expression of transcription factors of T regulator cells and the percent of different types of the immune cells in the blood (T-cell, B-cell, helper, and cytotoxic T-cells). Changing the expression of these markers under the influence of mature SPF female scent would help to understand the process of the formation of resistance to potential infections that the male may meet when searching for a sexual partner.

## 2. Materials and Methods

### 2.1. Housing Condition of Mice

All the procedures were conducted under Russian legislation according to Good Laboratory Practice (Directive of the Ministry of Health of the Russian Federation # 267, 19 June 2003), European Directive 2010/63/EU [36], and the European Convention for the protection of vertebrate animals used for scientific purposes. All procedures were approved by the bioethical committee (#18.6 14 October 2013).

BALB/cJNskrc mice were bred at the Center for Genetic Resources of Laboratory Animals of the Institute of Cytology and Genetics, Siberian Branch, Russian Academy of Sciences. The animal colonies were negative for all the pathogens listed in the annual FELASA 2014 recommendations from the time of their transfer from the Jackson Laboratory (2013) to the experiment’s completion. In accordance with the FELASA 2014 recommendations for surveillance, pathogen testing was conducted quarterly in sentinel mice that received both dirty bedding and water from colony mice for 3 months. Three days prior and during all experiments, male mice were housed singly in individually ventilated caging systems (Optimice^®^, Animal Care Systems, Centennial, CO, USA) to keep them isolated from the olfactory signals of other cages. Before the experiment, both female and male mice were housed in unisex groups (5 per cage) in the same room (breeding area) in open cages until 11 weeks old. Corticosterone levels were measured in both groups of animals with and without female scent (+FS and −FS) and the groups were socially isolated. As male isolation induces a rise in corticosterone levels, its effect was neglected. During the experiment, females were housed (5 per cage) in individually ventilated caging systems in a separate room from tested male mice. Mice were housed with autoclaved dust-free birch bedding, and they were provided sterile water and food ad libitum (Mouse Maintenance autoclavable, V1534-300, Sniff Spezialdiäten GmbH, Soest, Germany). The room had a photoperiod of 14-h light/10-h dark, a temperature of 22 ± 2 °C, and humidity of 40–50%.

### 2.2. Female Bedding Collection

Twenty-five mature female mice BALB/cJNskrc, 12 weeks old, in five cages, were used as female bedding donors to treat male mice. Female bedding that was soiled over 5 days was collected and added to the male cages every day, approximately 1 h after the lights were turned off. Examination of vaginal smears of all females two times per day for 5 days revealed at least 30% of female mice to be in estrus. The bedding sample from the female cages was contaminated by the urine and feces of female mice in estrus.

### 2.3. Experimental Groups

There were two independent experiments with male mice. In each experiment (n = 10 mice in the first experiment and n = 11 mice in the second experiment) 12-week-old BALB/cJNskrc mice were subjected to two types of scent treatment. In the first experiment, one group (5 male mice) received soiled bedding (8–10 g) from female cages (+FS group), and the other (5 male mice) received fresh and clean sawdust (−FS group). In the second experiment, M1 and M2 macrophages were assayed by cytometry using 7 males who received soiled bedding from female cages in the +FS group and 4 males who received fresh and clean sawdust in the −FS group. Soiled bedding or fresh flakes were placed into the right corner of male cages in a tea infuser (IKEA, stainless steel). The females’ bedding was added to the males’ cages and changed for a new portion every day for 7 days. The blood was collected from the retro-orbital sinus of both experimental groups without anesthesia as this can affect the hormonal level and immune cell count due to the glucocorticoid response. For 200 µL blood collection, the eyes of mice were treated with a drop of an ophthalmic anesthetic (0.5% proparacaine hydrochloride ophthalmic solution, Alcon Laboratories, Alcon-Couvreur N.V. S.A., Belgium). The blood was collected from the capillaries without anticoagulant for hormones assay and with heparin for cytometry of blood cells. After blood collection, the male mice were euthanized using CO_2_; their lungs (first experiment) were sampled in a sterile 1.5 mL plastic tube, immediately frozen in liquid nitrogen, and stored at −70 °C until the assays were performed. Bronchoalveolar lavages (second experiment) were collected in a sterile 15 mL plastic tube to analyze the M1 and M2 macrophage percentages.

To exclude the activation of the immunity of male mice by the components of microorganisms from soiled bedding, we used sib mice, which were housed in the same rooms and tested quarterly for the absence of specific pathogens. Before the experiment, the sibs of males and females were kept in the same room in open cages from birth until 11-weeks-old. Male and female sib mice were housed together until 3-weeks-old in the mother’s cage. Then, males and females were separated into unisex groups but housed in open cages in the same room for seven weeks. This approach to keeping sib mice from birth to puberty reduces fecal microbiota variability [37]. Three days before and during the experiment, the 12-week-old BALB/cJNskrc males were housed one per cage. The mice of the +FS group experienced daily exposure to soiled female bedding to provide the female scent. On the other hand, the −FS group was isolated from the female scent. The scent of the same and opposite sexes of mice causes sniffing and grooming behavior [38].

### 2.4. Sniffing Test to Female Scent

A new portion of scent stimuli was added into the males’ cages 1 h after the lights were turned off on the second day of scent treatment to test the sniffing reaction to the female scent by male mice. We recorded the time that male mice spent sniffing the tea infuser in both groups (+FS and −FS) over a 2 min period.

### 2.5. Cytometry of Blood Cells

For flow cytometry, red blood cells were lysed and washed twice as described previously [33]. Blood cells were stained with PE-anti-CD3ε (Hamster, clone 145-2C11), FITC-anti-CD4 (Rat IgG2b κ, clone GK1.5), PE/Cyanin7-anti-CD8a (Rat IgG2a κ, clone 53-6.7), PE-anti-CD3ε (Armenian Hamster IgG, clone 145-2C11), and FITC-anti-CD19 (Rat IgG2a κ, clone 6D5) anti-mouse antibodies (BioLegend, San Diego, CA, USA), PE-Cy7-CD45 (Rat169 IgG2b, k, clone 30-F11) for 120 min at 4 °C in the dark and then analyzed using a Guava easyCyte flow cytometer (Merck Millipore, Darmstadt, Germany). The populations of CD3+, CD19+, CD3+CD4+, and CD3+CD8+ cells were determined, and the number of cells was expressed as the total leukocyte count. The cell blood counts were made blind. For counting, cell aliquots of whole blood were stained with Türk’s solution (Cat. # 109277, Merck Millipore, Darmstadt, Germany), and the number of cells was determined using a counting chamber (Cat. # 12001711, Minimed, Bryansk, Russia) [39].

For macrophage isolation, the male mice were sacrificed, and bronchoalveolar lavage was collected. Male mice were euthanized by CO_2_ inhalation, and the lungs were washed three times with 5 mL of sterile ice-cold PBS. Suspension of bronchoalveolar cells was collected by catheters and centrifuged for 5 min at 1500 rpm. Cell pellets were resuspended in DMEM medium with 10% FBS and left to adhere in a 24-well culture plate (Corning) at 1 × 10^6^ cells per well for 1 h in CO_2_ incubator at 37 °C. Non-adherent cells were removed by washing twice with PBS medium. Adherent macrophages were used for further assays. Two hundred and fifty microliters of cell suspension were stained with FITC-F4/80 (Rat IgG2b, k, clone EMR1), PE-CD209a (Mouse IgG2c, clone MMD3), APC-anti-CD86 (Rat IgG2a, clone GL-1) (all BioLegend, San Diego, CA, USA) for 60 min at 4 °C in the dark and then samples were analyzed using a BD FACSCanto II Flow Cytometer.

### 2.6. Testosterone and Corticosterone Immunoassays

Testosterone and corticosterone concentrations in the blood serum were determined by a solid-phase enzyme immunoassay (ELISA). Testosterone-ELISA Kit (Chema Ltd., Moscow, Russia) and corticosterone-ELISA Kit (Enzo Lifescience, Lörrach, Germany) were used. The testosterone and the corticosterone assay’s sensitivities were 87 ng mL^−1^ and 27 ng mL^−1^, respectively. Serum testosterone and corticosterone levels were measured by an ELISA Kit, according to the manufacturer’s instructions.

### 2.7. Expression of Nos2, Arg1, and Foxp3 in Lungs

To measure the gene expression level of *Nos2*, *Arg1*, and *Foxp3*, total RNA was purified from lung samples using TRIzol reagent (Invitrogen, Waltham, MA, USA) according to the manufacturer’s recommendations (n = 5 for each group). Genomic DNA was removed from RNA samples using DNase I (ThermoScientific, Waltham, MA, USA) according to the manufacturer’s recommendations. RNA concentrations were determined with a NanoDrop 2000 spectrophotometer (ThermoScientific, Waltham, MA, USA). Between 5 and 7 µg of RNA was used for reverse transcription; cDNA synthesis was performed using M-MuLV reverse transcriptase (SibEnzyme, Novosibirsk, Russia), according to the manufacturer’s recommendations. A mix of random hexa-deoxyribonucleotide and Oligo-dT primers were used for reverse transcription. Upon the completion of DNA synthesis, the reaction was diluted with 4 volumes of deionized water. The real-time PCR reaction was prepared using a BioMaster HS-qPCR SYBR Blue (2×) (BioLabMix, Novosibirsk, Russia), 5 µL of cDNA, and 250 nM specific primers. Amplification and detection were performed using a CFX96 Touch™ Real-Time PCR Detection System (BioRad, Hercules, CA, USA). The relative level of gene expression was calculated using the CFX96 Touch™ software. Gene expression was normalized to Tubb5 (Tubulin, beta 5 class I) mRNA level and calculated as ∆Ct = 2^(Ct gene of interest mRNA − Ct Tubb5 gene of interest mRNA). Primer sequences used for real-time PCR analyses are shown in Table 1.

### 2.8. Statistical Analysis

Statistical analysis was performed with IBM^®^ SPSS^®^ Statistic version 23.0 software. The data were tested for normality using the Kolmogorov–Smirnov test. Normally distributed data were processed using Student’s *t*-test test. Not normally distributed data were processed using a Mann–Whitney U-test. All data are presented as the mean ± standard error.

## 3. Results

### 3.1. The Effect of Female Scent on Endocrine and Behavioral Factors of Males

In our study, the +FS male mice showed a significantly higher frequency of sniffing the female stimuli (65.1 ± 6.59 sec) than the −FS group (25.6 ± 9.38 sec; *p* = 0.01, Mann–Whitney U-test) (Figure 1c). Such an increased olfactory interest in the female scent ensured the animals’ contact with the stimuli presented. We did not have a control group of males that were subjected to the scent of other males. However, males were able to sniff their own scent. The reception by the main olfactory epithelium and the vomeronasal organ of olfactory stimuli involves the activation of the hypothalamus–pituitary–gonadal and glucocorticoid system [15,40]. The +FS group tended to have higher blood testosterone levels than the blood testosterone level of the −FS group, but this was not statistically significant (Mann–Whitney U-test, *p* = 0.08) (Figure 1a). Corticosterone levels did not differ between the two groups of males (Mann–Whitney U-test, *p* = 0.61) (Figure 1b). Thus, the +FS and −FS groups showed no difference in corticosterone or testosterone levels.

### 3.2. The Effect of Female Scent on the Number of White Blood Cells

Surprisingly, SPF female scent treatment of SPF male mice in the +FS group did not affect the total number of leukocytes (Mann–Whitney U-test, *p* = 0.71) and lymphocytes in the blood (Mann–Whitney U-test, *p* = 0.56) compared with mice in the −FS group (Figure 2a,b). However, after 7 days of sniffing the female scent, the percentage of T- and B-cells in the blood of BALB/c males were affected. The percentage of B-cells (CD19+) were higher in male mice of the +FS group compared to the −FS group (Mann–Whitney U-test, *p* = 0.04) (Figure 2c). At the same time, isolation from the female scent led to an increase in the percentage of T-cells (CD3+) in the −FS group (Mann–Whitney U-test, *p* = 0.04) (Figure 2d). However, the absolute values of T- (−FS: 1.21 ± 0.13; +FS: 1.37 ± 0.15 10^6^ cells) and B-cells (−FS: 1.47 ± 0.26; +FS: 1.08 ± 0.17 10^6^ cells) did not differ between groups (Mann–Whitney U-test, *p* = 0.60 and *p* = 0.25, respectively).

A decline in the level of specific immunoglobulins in male mice is known to be associated with a decrease in the percentage of T-cells [41]. T-helper cells are activated when presented with peptide antigens by molecules of class II of the major histocompatibility complex, which is expressed on the surface of antigen-presenting cells. T-helper cells’ main function is to assist other lymphocytes, including maturation of B-cells, plasma cells, and memory B-cells, and to activate cytotoxic T-cells and macrophages. Cytotoxic T-cells recognize antigens associated with the molecules of major histocompatibility complex I. The main function of cytotoxic T-cells is to destroy damaged cells. Nevertheless, the percentage of T-helpers, and the percentage of cytotoxic T-cells, did not differ between the +FS and −FS groups (Figure 2e,f). The absolute values of T-helper (−FS: 0.85 ± 0.08; +FS: 1.00 ± 0.10 10^6^ cells) and cytotoxic T-cells (−FS: 0.32 ± 0.04; +FS: 0.33 ± 0.04 10^6^ cells) did not differ between groups (Mann–Whitney U-test, *p* = 0.35 and, *p* = 0.75, respectively).

### 3.3. The Effect of Female Scent on the Expression of Nos2, Arg1, and Foxp3 in Lungs and the Percentage of M1 and M2 Macrophages in Bronchoalveolar lavage

The level of mRNA of the *Foxp3* gene, which is expressed in regulatory T-cells, was analyzed. It is known that regulatory T-cells are the main cells that express transcription factor *Foxp3*, which regulates the transcription of genes responsible for T-cell differentiation and the expression of cytokines and other factors involved in suppressing the immune response [42]. Seven days of sniffing the female scent resulted in a drop in *Foxp3* mRNA in male mice’s lungs, but this was not statistically distinguished (Mann–Whitney U-test, *p* = 0.07). T-regulatory cells are the central regulators of the immune response. Their main function is to control the strength and duration of the immune response by regulating T-effector cells’ function (T-helper and T-cytotoxic cells). T-regulatory cells express the transcription factor gene *Foxp3*. Antigen-activated T-cells act on receptors on regulatory T-cells, warning them that there is high T-cell activity in the region, and they elicit an inhibitory response against them, a negative feedback loop to avoid overreaction [43]. If a real infection is present, regulatory factors suppress the hyperinflammatory reaction. It has been proposed that sniffing the SPF female scent may affect the main metabolites of the macrophage types, which may be involved in T-cell polarization (type 1 and 2 T-helper cells). It is well known that the *Nos2* gene is expressed by activated M1 macrophages and *Arg1* by M2 macrophages. *Nos2* and *Arg1* gene expression differed in male mice of the +FS group and the −FS group. The expression of the *Nos2* gene in the lungs differed by 20 times, and *Arg1* by five times between the +FS and the −FS groups. The mRNA levels of *Nos2* and *Arg1* were significantly higher in males exposed to the female scent (Mann–Whitney U-test, *p* = 0.009 and *p* = 0.008, respectively) (Figure 3a,b).

Since we found an increase in the expression of *Arg1* and *Nos2* genes, which are the main markers of M1 and M2 macrophages, we decided to check how the percentage of macrophages in the bronchoalveolar lavage differed between the +FS and −FS groups. The percentage of M1 macrophages was higher in the +FS group than in the −FS group (Mann–Whitney U-test, *p* = 0.003) (Figure 3f). The percentage of M2 macrophages did not differ between the +FS and −FS groups (Mann–Whitney U-test, *p* = 0.12) (Figure 3d). However, the percentage of boarding M1/M2 macrophages in the +FS group was lower in than the −FS group (Mann–Whitney U-test, *p* = 0.02) (Figure 3e). These data indicate that SPF mice’s female scent increased the percentage of M1 macrophages in the bronchoalveolar lavage of male mice.

Thus, seven days of sniffing the SPF female scent increased the mRNA of metabolites of macrophages of both types in the lungs of male mice. As such, this activation could be useful to protect males against pathogenic infection.

## 4. Discussion

The previously shown redistribution of leukocytes between the blood and lungs was due to the reaction of the gonadal system to the stimuli of the scent of conventional status females [7,8,9]. White blood cells accumulate in the lungs and upper respiratory tract to protect the animals from potential bacterial and viral threats caused by olfactory searching for sexual partners [13]. In the absence of specific pathogens, the female scent did not change the testosterone level or the number of total leukocytes in the male mice (Figure 1a,b). The changes in the male innate immunity of the lung, mediated by the sniffing of SPF female scent, were not associated with activation of the gonadal or adrenal system in male SPF mice. It can be assumed that males in presence of female scent have a higher sniffing activity, that leads to an increased inhalation of bacterial-associated metabolites and odor molecules. Female bacterial-associated metabolites and odor molecules can stimulate receptors on the immune cells in the lungs and affect the immunity of mucosal and systemic responses. In contrast males living in isolation from female stimuli do not demonstrate similar effect.

We supposed that the female scent and bacterial patterns from soiled bedding modulate immunity. Previously it has been shown that the scent of female mice of conventional status causes a redistribution of immune cells between the lungs and blood, which contributes to the resistance of male mice to viral infection [8,9,13]. Moreover, the sniffing of the female scent has been shown to decrease specific humoral immunity in males by reducing the number of antibodies producing cells in the spleen [5,6]. The scent of females activates innate immunity in males. Simultaneously, the specific humoral immunity decreases, possibly due to the activation of T regulatory cells.

The redistribution between the functioning of the innate and humoral (specific) immunity in response to the female scent indicates the innate protective mechanisms during the search for sexual partners in rodents [6,7]. All previous studies were performed on mice that have specific pathogens (conventional status). Pathogen-associated molecular patterns from soiled bedding of conventional female mice could activate innate immunity and induce endocrine responses. Thus, the effect of the female scent is absent or combined with the reception of pathogen-associated molecular patterns. In this study, we used animals free of specific pathogens (SPF mice), thereby directly assessing the impact of the female odor molecules or the bacterial-associated metabolites of the gut’s normal microbial content on the redistribution of immune cells in the blood and the activation of innate immunity in the lungs. So, soiled bedding from SPF female mice activated innate immunity, but did not induce an endocrine response.

T-cell dependent antibody production in spleen is associated with type 2 T-helper cells. The percentage of T-cells (CD3-positive) decreased in the +FS group of mice (Figure 2d). Decline of T-cells in the +FS mice could affect on the specific antibody production in the spleen. However, it has been shown that the scent of SPF female mice does not reduce antibody production of male mice [44]. In the case of pathogen presence, metabolic host resources (ATP, peptides, etc.) are redistributed between the reproductive system and the immune system. There is no redistribution between gonadal and immune systems if mice are free of pathogen. Those mice will not experience significant changes in their testosterone levels or antibody responses because the resources become sufficient for the immune response and the activation of the reproductive system. However, contact with scent of SPF female mice is sufficient to shift the balance of T- and B-cells (Figure 2c,d). Whether odor molecules or bacterial-associated metabolites of the soiled bedding cause this shift remains unclear. It requires further research using transgenic mice lacking the receptors involved in recognizing odor molecules or bacterial-associated metabolites.

One of the possible methods of cell activation associated with female scent is activating a number of the receptors of immune cells. During the sniffing of soiled bedding, volatile molecules and bacterial components enter the recipient mucosa and interact with receptors on epithelial and immune cells [25,43,45,46]. In this regard, we propose that sniffing the female scent can affect the main metabolites of two types of macrophages that can polarize T-cells. Indeed, sniffing the female scent stimulated M1 macrophages in the lungs, reflected in changes in *Nos2* gene expression in the lung and in the percentage of CD209-CD86+ in F4/80-positive M1 macrophages. Active macrophages increase the resistance to potentially infectious agents. Thus, increased *Nos2* expression leads to the production of free NO radicals, which protect against both viruses and bacteria [47]. Moreover, the level of Arg1 gene mRNA was also higher in the +FS group. Arginase 1 indicates the launch of anti-inflammatory mechanisms of the immune system [48]. Epithelial cells as well as macrophage and dendritic cells can express Arginase 1 [49]. Thus, the rise in Arg1 gene mRNA levels in male mice in the +FS group could be related to macrophages and epithelial cells, as this helps to maintain a balance between inflammatory and anti-inflammatory processes. Most likely, in the presence of the female scent macrophage activity in the lung provides resistance to the influenza virus in male mice [13].

The female scent decreased the percentage of T-cells and activated macrophages. We suggested that immunity should trigger a regulatory mechanism, as in various infections. However, we did not find significant changes in the mRNA expression of the Foxp3 gene in T regulatory cells. Since we did not find changes in *Foxp3* gene expression, T-cells’ regulatory function in male mice sniffing of SPF female scent was not triggered. Infection in conventional status mice activated an acute inflammatory process, which is controlled by T regulatory cells [42,43]. For example, the introduction of bacterial components (e.g., lipopolysaccharides) into the lung causes pro-inflammatory reactions and increases IL-1beta level [8,50,51]. T-regulatory cells play an important role in reducing inflammatory responses, which normalize and relieve acute inflammation and pro-inflammatory cytokines in the lung [52]. Contact with the female scent did not cause an increase in Il-1beta in lung tissue, in contrast to the lipopolysaccharide of *Escherichia coli* [52]. The absence of inflammation and many T-regulatory cells after female scent exposure may indicate the activation of the protective mechanisms of innate immunity. Sexual behavior provokes sniffing activities among males, which leads to an increased risk of infection. Therefore, the activation of barrier immunity is necessary as an adaptation to an increased risk of infection. Indeed, the female scent promotes innate immunity—the migration of macrophages and changes in T- and B-cell ratios—but without acute inflammatory reactions [8,52]. Still the mechanisms of this phenomenon are not fully understood.

## 5. Conclusions

This study demonstrates the activation of M1 type macrophages in the lungs of male mice upon exposure to the SPF female scent. Such a reaction in the immune system of the lung could contribute to resistance to the viral and bacterial infections that the male could potentially encounter during breeding behavior.

## Figures and Tables

**Figure 1 animals-11-01756-f001:**
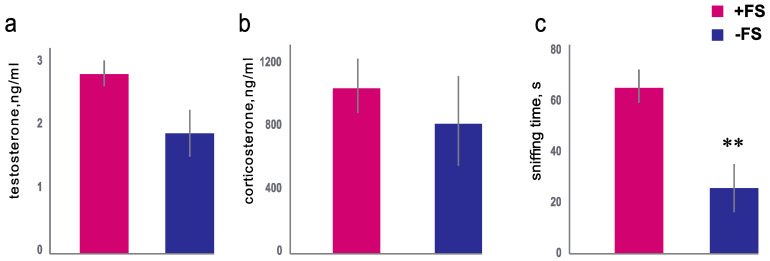
Behavioral and endocrine responses of male mice to SPF female scent. (**a**) The testosterone level (ng/mL) in the blood of the males in +FS group (group of males received bedding of females) (red column) or in −FS group (group isolated from female scent) (blue column). (**b**) The corticosterone level (ng/mL) in the blood of males in +FS group (red column) or in −FS group (blue column). (**c**) Frequency of sniffing of +FS group (red column) or −FS group (blue column) **-*p* < 0.01 Mann–Whitney U-test. Data are presented as the mean ± standard error.

**Figure 2 animals-11-01756-f002:**
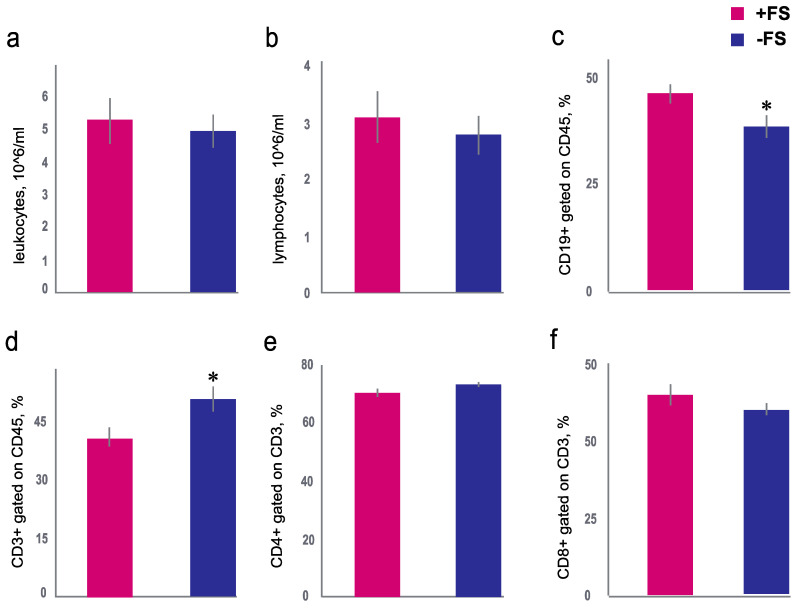
Effect of exposure to female scent on the blood leukocytes of male mice. (**a**) The number of leukocytes in the blood for +FS group (red column) and −FS group (blue column). (**b**) The number of lymphocytes in the blood for +FS group (red column) and −FS group (blue column). (**c**) The percentage of B cells (CD19) in the blood for +FS group (red column) and −FS group (blue column). (**d**) The percentage of T-cells (CD3) in the blood for +FS group (red column) and −FS group (blue column). (**e**) The percentage of T helper cells (CD4+) in the blood for +FS group (red column) and −FS group (blue column). (**f**) The percentage of T cytotoxic cells (CD8+) in the blood for +FS group (red column) and −FS group (blue column). *-*p* < 0.05 Mann–Whitney U-test. Data are presented as the mean ± standard error.

**Figure 3 animals-11-01756-f003:**
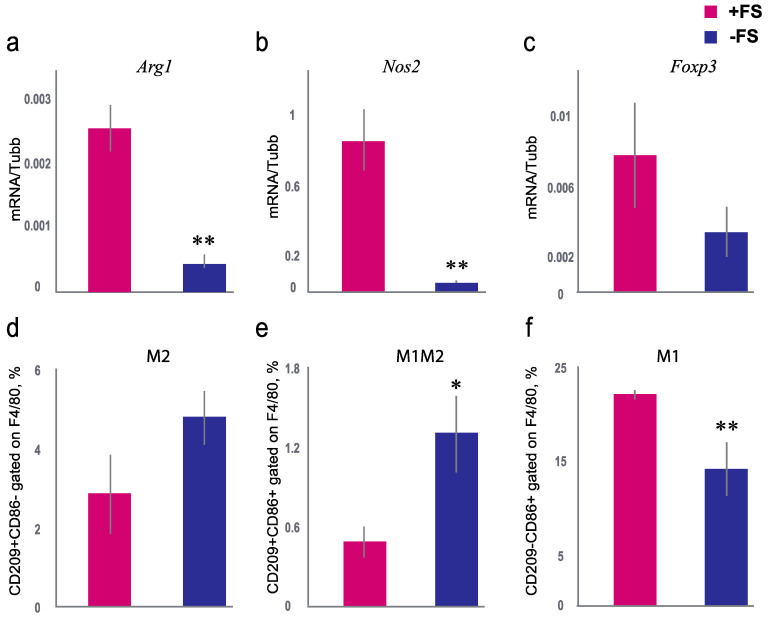
Effect of exposure to female scent on the expression of *Arg1*, *Nos2*, and *Foxp3* genes in the lung of male mice and percentage of macrophages of M1 and M2 types in broncho-alveolar lavage. (**a**) The expression of *Arg1* level in the lung for the +FS group (red column) and −FS group (blue column) measured by RT-PCR. (**b**) The expression of *Nos2* mRNA level in the lung for the +FS group (red column) and −FS group (blue column) measured by RT-PCR. (**c**) The expression of *Foxp3* mRNA level in the lung for the +FS group (red column) and −FS group (blue column) measured by RT-PCR. The mRNA level of *Tubb5* (Tubulin, beta 5 class I) used as an internal standard. (**d**) The percentage of M2 type macrophages in broncho-alveolar lavage for +FS group (red column) and −FS group (blue column). (**e**) The percentage of boarding M1M2 type macrophages in broncho-alveolar lavage for +FS group (red column) and −FS group (blue column). (**f**) The percentage of M1 type macrophages in broncho-alveolar lavage for +FS group (red column) and −FS group (blue column). *-*p* < 0.05, **-*p* < 0.01 Mann–Whitney U-test. Data are presented as the mean ± standard error.

**Table 1 animals-11-01756-t001:** Primers used for the study.

Primer Name	Sequence 5′->3′	Target
Nos2_F	CAGGGTCACAACTTTACAGGGA	Mouse *Nos2*
Nos2_R	CACTTCTGCTCCAAATCCAACG	
Arg1_F	TCACCTGAGCTTTGATGTCGA	Mouse *Arg1*
Arg1_R	TGAAAGGAGCCCTGTCTTGTA	Mouse *Foxp3*
Foxp3_F	AGAGTTTCTCAAGCACTGCCA	
Foxp3_R	TCCCAGCTTCTCCTTTTCCA	
Tubb_F	TGAAGCCACAGGTGGCAAGTAT	Mouse *Tubb5*
Tubb_R	CCAGACTGACCGAAAACGAAGT	

## Data Availability

No new data were created or analyzed in this study. Data sharing is not applicable to this article.

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
