# Peer review of "Female Scent Activated Expression of Arginase1 and Inducible NO-Synthetase in Lung of BALB/c Male Mice"

_animals, 2021, doi:10.3390/ani11061756_

Round 1
Reviewer 1 Report
Summary: The manuscript of Gvozdeva et al. “Female scent activated expression of Arginase1 and inducible NO-synthetase in lung of BALB/c male mice” investigates the effect of estrus female scent on the immune system of male mice. The authors describe the modification of the percentage of different leukocytes population in peripheral blood and the upregulation of arg1 and iNOS gene expression in lung.
General comments:
The English of the manuscript has been really improved except for the discussion section, that still need English editing. Some sentences are still difficult to understand. Material and method section has been improved and the new experiments improved the experimental design. The authors reply that species-specific pathogen means SPF, but this is not correct. SPF=specific pathogen free, please delete “species-” before “specific” in all the manuscript.
Specific comments
Abstract:
Line 34: delete “activating” replace with “activation”
Introduction:
Lines 50-56: please add references.
Line 57: add “,” after “lung”
M&M
Lines 136-140: authors should better explain the number of experiments and the number of animal per group in each experiments; it is not clear how they obtain 7 samples from groups of 5 animals…..
Line 147: how the authors isolated the leukocytes from blood samples without anticoagulant?
Line 205: the formula of Delta Ct is not correct. DCt=Ct of target gene-Ct of housekeeping gene.
Results
Line 224: what the authors means by “new microrganisms”?
Lines 260-264: the sentences should be moved to discussion section.
Lines 310-315: the sentences is not clear and in any case it should be moved to discussion section.
Lines 326-327: the sentences is not clear.
Lines 330-332: the sentence is not clear. What the authors means by “increased expression of metabolites”? Metabolites are not quantified by gene expression!
Discussion
Lines 352-355: the sentence is not clear. Do they mean that males in presence of female scent have a higher sniffing activity, that lead to an increased inhalation of bacterial-associated metabolites or molecules? The term “bacterial pattern” is not correct; does it stay for “bacterial-associated molecular patterns”?
Lines 272-273: “antibody producing cells” of which district (spleen, lymph nodes, MALT)?
Line 360: delete “antibodies forming cells” replace with “antibody producing cells”.
Line 361: what is “non specific barrier immunity”? Do the authors mean innate immunity? Or mucosal immunity?
Lines 363-365: the sentence seems to be the opposite of the results; the innate immunity seems to be activate in +FS group.
Line 367: what is “pathogenic patterns”? Pathogen-associated molecular patterns (PAMPs)? But the hormonal response didn’t show any difference between the two groups.
Line 371: “barrier immunity”? innate or mucosal?
Lines 372-373 and 379-380: this statement is not in line with the results where the total number of T cells was not affected by the stimulation of males with FS.
Author Response
Dear Reviewer,
We are grateful to the reviewer for analyzing our manuscript. Below we provide responses to all reviewer comments.
Summary: The manuscript of Gvozdeva et al. “Female scent activated expression of Arginase1 and inducible NO-synthetase in lung of BALB/c male mice” investigates the effect of estrus female scent on the immune system of male mice. The authors describe the modification of the percentage of different leukocytes population in peripheral blood and the upregulation of arg1 and iNOS gene expression in lung.
General comments:
The English of the manuscript has been really improved except for the discussion section, that still need English editing. Some sentences are still difficult to understand. Material and method section has been improved and the new experiments improved the experimental design. The authors reply that species-specific pathogen means SPF, but this is not correct. SPF=specific pathogen free, please delete “species-” before “specific” in all the manuscript.
Response:
We’ve made changes to the discussion section. We’ve corrected style of sentences. We’ve removed duplicate sentences and transferred some of the sentences from the results section.
We’ve send our discussion to English native speaker and she’ve corrected its.
Specific comments
Abstract:
Line 34: delete “activating” replace with “activation”
- replaced
Introduction:
Lines 50-56: please add references.
- corrected, lines 50-56 included three references
Line 57: add “,” after “lung”
- added “,” after “lung”
M&M
Lines 136-140: authors should better explain the number of experiments and the number of animal per group in each experiments; it is not clear how they obtain 7 samples from groups of 5 animals….
- Included information about two experiments and added how many mice were in each experimental groups for each experiments.
Line 147: how the authors isolated the leukocytes from blood samples without anticoagulant?
- Indeed, we collected blood for hormones assay without anticoagulation and for cytometry of blood cells with heparin. Corrected, lines 149-150
Line 205: the formula of Delta Ct is not correct. DCt=Ct of target gene-Ct of housekeeping gene.
- corrected, line 208
Results
Line 224: what the authors means by “new microrganisms”?
- corrected, line 227. Delated “new” from this sentence
Lines 260-264: the sentences should be moved to discussion section.
- corrected, lines 353-358. Moved to discussion section
Lines 310-315: the sentences is not clear and in any case it should be moved to discussion section.
- corrected, Rewrote and moved to the discussion section.
Lines 326-327: the sentences is not clear.
- Rewrote, line 433-434 “However, the percentage of boarding M1/M2 macrophages in the +FS group was lower than the −FS group”
Lines 330-332: the sentence is not clear. What the authors means by “increased expression of metabolites”? Metabolites are not quantified by gene expression!
- Changed to “ increased the mRNA of metabolites of macrophages”
Discussion
Lines 352-355: the sentence is not clear. Do they mean that males in presence of female scent have a higher sniffing activity, that lead to an increased inhalation of bacterial-associated metabolites or molecules? The term “bacterial pattern” is not correct; does it stay for “bacterial-associated molecular patterns”?
Rewrite:
- Rewrote, lines 360-363. It can be assumed that males in presence of female scent have a higher sniffing activity, that lead to an increased inhalation of bacterial-associated metabolites and odor molecules, in contrast to males living in isolation from female stimuli, as bacterial-associated metabolites and odor molecules can stimulate receptors on the immune cells in the lungs and affect the immunity of mucosal and systemic responses.
Lines 272-273: “antibody producing cells” of which district (spleen, lymph nodes, MALT)?
- Added “in spleen”
Line 360: delete “antibodies forming cells” replace with “antibody producing cells”.
- replaced
Line 361: what is “non specific barrier immunity”? Do the authors mean innate immunity? Or mucosal immunity?
- corrected, lines 371-372. We mean innate immunity.
Lines 363-365: the sentence seems to be the opposite of the results; the innate immunity seems to be activate in +FS group.
- The sentence was rewritten
Line 367: what is “pathogenic patterns”? Pathogen-associated molecular patterns (PAMPs)? But the hormonal response didn’t show any difference between the two groups.
- corrected, line 380. “Pathogenic patterns” were replace to “Pathogen-associated molecular patterns from soiled bedding of conventional female mice”. we also included sentence “So, soiled bedding from SPF female mice activated innate immunity, but not induced of endocrine response.”
Line 371: “barrier immunity”? innate or mucosal?
- replaced to “innate”
Lines 372-373 and 379-380: this statement is not in line with the results where the total number of T cells was not affected by the stimulation of males with FS.
- Rewrote sentence and added a link to Fugure2d and 2c.

Reviewer 2 Report
No more comments
Author Response
Dear Reviewer,
We are grateful to the reviewer for analyzing our manuscript. Below we provide responses to all reviewer comments.
Summary: The manuscript of Gvozdeva et al. “Female scent activated expression of Arginase1 and inducible NO-synthetase in lung of BALB/c male mice” investigates the effect of estrus female scent on the immune system of male mice. The authors describe the modification of the percentage of different leukocytes population in peripheral blood and the upregulation of arg1 and iNOS gene expression in lung.
General comments:
The English of the manuscript has been really improved except for the discussion section, that still need English editing. Some sentences are still difficult to understand. Material and method section has been improved and the new experiments improved the experimental design. The authors reply that species-specific pathogen means SPF, but this is not correct. SPF=specific pathogen free, please delete “species-” before “specific” in all the manuscript.
Response:
We’ve made changes to the discussion section. We’ve corrected style of sentences. We’ve removed duplicate sentences and transferred some of the sentences from the results section. We’ve send our discussion to English native speaker and she’ve corrected its.
Specific comments
Abstract:
Line 34: delete “activating” replace with “activation”
- replaced
Introduction:
Lines 50-56: please add references.
- corrected, lines 50-56 included three references
Line 57: add “,” after “lung”
- added “,” after “lung”
M&M
Lines 136-140: authors should better explain the number of experiments and the number of animal per group in each experiments; it is not clear how they obtain 7 samples from groups of 5 animals….
- Included information about two experiments and added how many mice were in each experimental groups for each experiments.
Line 147: how the authors isolated the leukocytes from blood samples without anticoagulant?
- Indeed, we collected blood for hormones assay without anticoagulation and for cytometry of blood cells with heparin. Corrected, lines 149-150
Line 205: the formula of Delta Ct is not correct. DCt=Ct of target gene-Ct of housekeeping gene.
- corrected, line 208
Results
Line 224: what the authors means by “new microrganisms”?
- corrected, line 227. Delated “new” from this sentence
Lines 260-264: the sentences should be moved to discussion section.
- corrected, lines 353-358. Moved to discussion section
Lines 310-315: the sentences is not clear and in any case it should be moved to discussion section.
- corrected, Rewrote and moved to the discussion section.
Lines 326-327: the sentences is not clear.
- Rewrote, line 433-434 “However, the percentage of boarding M1/M2 macrophages in the +FS group was lower than the −FS group”
Lines 330-332: the sentence is not clear. What the authors means by “increased expression of metabolites”? Metabolites are not quantified by gene expression!
- Changed to “ increased the mRNA of metabolites of macrophages”
Discussion
Lines 352-355: the sentence is not clear. Do they mean that males in presence of female scent have a higher sniffing activity, that lead to an increased inhalation of bacterial-associated metabolites or molecules? The term “bacterial pattern” is not correct; does it stay for “bacterial-associated molecular patterns”?
Rewrite:
- Rewrote, lines 360-363. It can be assumed that males in presence of female scent have a higher sniffing activity, that lead to an increased inhalation of bacterial-associated metabolites and odor molecules, in contrast to males living in isolation from female stimuli, as bacterial-associated metabolites and odor molecules can stimulate receptors on the immune cells in the lungs and affect the immunity of mucosal and systemic responses.
Lines 272-273: “antibody producing cells” of which district (spleen, lymph nodes, MALT)?
- Added “in spleen”
Line 360: delete “antibodies forming cells” replace with “antibody producing cells”.
- replaced
Line 361: what is “non specific barrier immunity”? Do the authors mean innate immunity? Or mucosal immunity?
- corrected, lines 371-372. We mean innate immunity.
Lines 363-365: the sentence seems to be the opposite of the results; the innate immunity seems to be activate in +FS group.
- The sentence was rewritten
Line 367: what is “pathogenic patterns”? Pathogen-associated molecular patterns (PAMPs)? But the hormonal response didn’t show any difference between the two groups.
- corrected, line 380. “Pathogenic patterns” were replace to “Pathogen-associated molecular patterns from soiled bedding of conventional female mice”. we also included sentence “So, soiled bedding from SPF female mice activated innate immunity, but not induced of endocrine response.”
Line 371: “barrier immunity”? innate or mucosal?
- replaced to “innate”
Lines 372-373 and 379-380: this statement is not in line with the results where the total number of T cells was not affected by the stimulation of males with FS.
- Rewrote sentence and added a link to Fugure2d and 2c.

This manuscript is a resubmission of an earlier submission. The following is a list of the peer review reports and author responses from that submission.
Round 1
Reviewer 1 Report
Comments:
- On Figure 1: label "testosterone", "corticosterone", and "frequency of sniffing" on the figure. Also label p value on the figure.
- On Figure 2: label p value on figure and legend
- On figure 3: label p value on figure and legend
- measure Arg1, iNOS, and Foxp3 protein levels by Western Blot analysis
- Please explain why study iNOS, Arg1, and Foxp3 specifically.
Reviewer 2 Report
Summary: The manuscript of Gvozdeva et al. “Female scent activated expression of Arginase1 and inducible NO-synthetase in lung of BALB/c male mice” investigates the effect of estrus female scent on the immune system of male mice. The authors describe the modification of the percentage of different leukocytes population in peripheral blood and the upregulation of arg1 and iNOS gene expression in lung.
General comments:
The topic of the manuscript is original and novel, but the experimental design presents some important bias. The manuscript need important English editing, some sentences are difficult to understand due to the poor English. Material and method section need more information to understand the experimental design. Conclusions are not supported by the results obtained by the authors. One of the main problem is that the gene expression was made on lung, and not on leukocyte populations from lungs. Immunohistochemistry or cytofluorimetric analysis on disaggregated cell from lung should be more appropriate. It is not clear the term "species-specific pathogens"; mice in the animal facility can be infected not only by species-specific pathogens, but also by non species-specific pathogens such as S. aureus; may the authors mean SPF mice, that is a different concept.
Specific comments
Introduction:
The authors should better describe the role of the different leukocyte populations in the lung and the characteristic of macrophages differently activated (M1 and M2).
Lines 47-50: I couldn’t find the reference n 7. But I can’t understand how the authors excluded the presence of PAMPs in the female scent (infected with pathogens) applied to males. If PAMPs were present how the authors can distinguish the activity of female scent vs the PAMPs activity on the immune system?
M&M
Lines 100: authors should take into account the fact that the isolation of one animal from the group can cause stress, and stress can affect the immune system and its responses. Did the authors measure the cortisol of the animals?
Line 109: the authors should mention the new European legislation on protection of laboratory animals.
Line 126: it is not clear. Blood was collected only from one experimental group?
Line 128: how many blood was collected? How was it collected (capillary, eye dislocation)? Which anticoagulant?
For how many days the bedding from female was added in the cages of the males?
Line 134: only 2 min is it sufficient?
Line 145: were the cell blood counts made in blind? The CD45 staining is missing.
Line 168: it is not clear which method was used to calculate gene expression.
Line 177: did the authors analyse the data distribution with Normality test? Mann-Whitney test is a non parametric test to use on non parametric data.
Results
Line 183: what age were the animals weaned? Leave males and females sibs together until 11 weeks is too much. Mice can reach puberty at 6-8 weeks.
Line187: the sentence is not clear.
Line 191: an important control group should be males subjected to male scent.
Lines 194-196: the sentence is not clear. Compared to?
Lines 198-199: the sentence is not clear.
Figure 1: the name of the groups should be modified because they are not clear.
Lines 215-216: the sentence is not clear.
Lines 218-220: the total counts are not shown. There are differences in total count? In the text these data should be present.
Line 219-220: where is the graph on the increase of T cells?
Lines 221-225: the sentence is not clear and some statement are not correct (T helper are not antigen presenting cells, but are activated by APCs to activate B cells; macrophages and DCs are antigen presenting cells, not T helper; T cytotoxix cells do not present intracellular antigens).
Fig. 2. Please insert Standard deviation or standard error bars. The figure caption mention blu and red dots…but in the fugure the dots are black and white.
Line 236: there are study that suggest the expression of FoxP3 also in other cells (B, macrophages, epithelial). It is not clear why the authors investigate the expression of FoxP3.
Lines 238-240: the sentence is not clear. The Hypothesis is that the female scent activates the immune system of the males?
Lines 240-241: the sentence is not clear. Why the anti-inflammatory response should be significative?
Fig. 3. . Please insert Standard deviation or standard error bars. The figure caption mention blu and red dots…but in the fugure the dots are black and white.
Discussion
Lines 263-268: why the authors speak about “upper airways” if they analysed the lungs? Do they refer to studies of other researcher? In the last case please add the references.
Lines 272-273: “antibody producing cells” of which district (spleen, lymphnodes, MALT)?
Lines 273-275: the sentence is not clear. What is the meaning of “redistribution”? What is the meaning of “non-specific and specific immunity”? The sentence is in contradiction with previous sentences that mentioned a reduction of antibodies.
Lines 282-283: This is in contradiction with the result section were the T are shown to increase.
Lines 283-285: the sentence is not clear.
Lines 285-286: redistribution of what? Energy, cells…..
Lines 299-304: it is not clear the discussion about these receptors in the context of the present study.
Lines 323-325: the sentence is not clear.